# Extension of the Theory of Planned Behavior (TPB) to Predict Patterns of Marijuana Use among Young Iranian Adults

**DOI:** 10.3390/ijerph17061981

**Published:** 2020-03-17

**Authors:** Farzad Jalilian, Mehdi Mirzaei-Alavijeh, Mohammad Ahmadpanah, Shayan Mostafaei, Mehdi Kargar, Razieh Pirouzeh, Dena Sadeghi Bahmani, Serge Brand

**Affiliations:** 1Lifestyle Modification Research Center, Imam Reza Hospital, Kermanshah University of Medical Sciences, Kermanshah 6719851351, Iran; f_jalilian@yahoo.com; 2Social Development & Health Promotion Research Center, Health Institute, Kermanshah University of Medical Sciences, Kermanshah 6719851351, Iran; mehdimirzaiea@yahoo.com (M.M.-A.); r_pirouzeh@yahoo.com (R.P.); 3Behavioral Disorders and Substance Abuse Research Center, Hamadan University of Medical Sciences, Hamadan 6719851351, Iran; m1ahmad2000@gmail.com; 4Medical Biology Research Center Research, Kermanshah University of Medical Sciences, Kermanshah 6719851351, Iran; Shayanmostafaee@yahoo.com; 5Department of Health Promotion, Faculty of Health, Shiraz University of Medical Sciences, Shiraz 7134814336, Iran; kargarmehdi4@gmail.com; 6Psychiatric Clinics of the University of Basel, Center for Affective, Stress und Sleep Disorders, University of Basel, 4002 Basel, Switzerland; dena.sadeghibahmani@upk.ch; 7Departments of Physical Therapy, University of Alabama at Birmingham, Birmingham, AL 35209, USA; 8Sleep Disorders Research Center, Kermanshah University of Medical Sciences (KUMS), Kermanshah 6719851115, Iran; 9Substance Abuse Prevention Research Center, Health Institute, Kermanshah University of Medical Sciences (KUMS), Kermanshah 6719851115, Iran; 10School of Medicine, Tehran University of Medical Sciences, Tehran 1416753955, Iran; 11Department of Sport, Exercise and Health, Division of Sport and Psychosocial Health, University of Basel, 4053 Basel, Switzerland

**Keywords:** marijuana use, theory of planned behavior, young adults, problem-solving skills, self-efficacy

## Abstract

*Background*: Marijuana use is increasing among adolescents and young adults. Long-term marijuana use magnifies the risk of a wide variety of behavioral, cognitive-emotional, and neurological problems, and can be a gateway to use of other drugs. In the present study, we investigated the cognitive-emotional and behavioral predictors of marijuana use. To this end, young Iranian adults answered questions based on an extended Theory of Planned Behavior (TPB) and related it to marijuana use. We hypothesized that cognitive-emotional and behavioral factors would predict intention to use marijuana, and that this, in turn, would predict actual consumption. *Methods*: A total of 166 young Iranian adults (mean age: 20.51 years; 15.7% females) attending a walk-in center for drug use took part in this cross-sectional study. Participants completed questionnaires covering sociodemographic information, frequency of marijuana use per week, along with questionnaires assessing the following dimensions of the TPB: attitude towards marijuana use, subjective norms, self-efficacy to resist marijuana use, environmental constraints, problem-solving skills, and behavioral intention for marijuana use. *Results*: Mean marijuana use was found to be 4.6 times/week. Attitude towards marijuana use, subjective norms, environmental constraints, and behavioral intention to use marijuana were positively correlated to each other and with marijuana use/week. In contrast, higher self-efficacy and problem-solving skills were associated with lower marijuana use/week. The multiple regression analysis showed that a positive attitude to marijuana use, lower self-efficacy in resisting its use, higher behavioral intention, and poorer problem-solving skills predicted actual use. *Conclusion*: The pattern of results suggests that dimensions of TPB can explain marijuana use among young Iranian adults self-admitted to a walk-in center for drug use. Specifically, poor problem-solving skills, low self-efficacy in resisting marijuana use, and positive labelling of its use appeared to be the best predictors of actual use. It follows that prevention programs aimed at improving problem-solving skills and raising self-efficacy, along with educational interventions aimed at highlighting the negative effects of marijuana might decrease the risk of its use among young adults in Iran.

## 1. Introduction

Marijuana is the drug most commonly abused by teenagers and young adults worldwide [1]. Typically, marijuana use among adolescents and young adults is related to recreational/leisure time rather than medical needs. In the USA, Rubinstein et al. [2] reported that 79.5% of a sample of 13–17-year-olds had smoked marijuana in the past 30 days. Johnson et al. [3] estimated that the prevalence rate for marijuana use in the past 30 days among adolescents was about 22.5%, and appeared to have remained stable from 1999 to 2013. 

According to the European Drug Report [4], prevalence rates for marijuana use among individuals aged 15 to 34 years ranged from 3.5% (Hungary) to 21.5% (France) in 2018. The same report estimated that more than one quarter of Europeans aged 15 to 64 years had tried cannabis during their lifetime. In the United States, the prevalence rate for use doubled from 4.1% in 2002/3 to 9.5% in 2012/13. Over the same period, prevalence rates for marijuana use disorder decreased from 35.6% to 30.5% (see Hasin et al. [5] for further details). Despite this decrease, in absolute figures, 30.5% out of 9.5% represents more marijuana users that 35.6% out of 4.1%. Next, following the study of Hasin et al. in 2012 [5], three in 10 marijuana users suffered from marijuana use disorder, while Hall and Degenhardt [6] concluded that about 9% of marijuana users were also addicted.

With regard to Iran, following Ghiabi et al. [7], the government is currently reviewing cannabis and opium regulations. The review could result in legalisation of drug consumption through a state-supervised system. While the analgesic and appetite-increasing effects of marijuana are acknowledged for medical reasons, the recreational use of marijuana (or opium) is not likely to be legalised. 

With regard to the illicit use of marijuana, in the north and northwestern part of the country, the prevalence rate for its use among high school students has been estimated at 22.2% [8]. Nazarzadeh et al. [9] reported in their systematic review a prevalence rate of 4% for cannabis abuse among Iranian high school and college students. Similarly, Momtazi and Rawson [10] reported that illicit substance use was a serious health problem among Iranian high school students: 4.4% to 12.8% reported daily tobacco use, and about 9.9% reported alcohol use at least once in their lifetime. Sajjadi et al. [11] showed that, for university students, having close friends with high-risk behavior (use of illicit drugs, such as alcohol or cannabis, and medications; extramarital heterosexual intercourse) was associated with individuals’ own high-risk behavior, such as drug use. In terms of the Theory of Planned Behavior (TPB), these results support the idea that there is an influence of social and subjective norms on individuals’ intentions to engage in high-risk behaviors. 

Typical adverse health effects of regular and heavy marijuana use are psychiatric problems, such as depression, anxiety, suicidality, psychosis [1], and schizophrenia [12,13,14]. Marijuana use has numerous consequences, including impaired respiratory function and cardiovascular disease [6], increase in myocardial infarction, and stroke prevalence [15], along with neural connectivity impairment, and hippocampus activity reduction [15]. To illustrate further, Hasin et al. [5] listed the following somatic, cognitive, behavioral, and emotional consequences: cognitive decline, psychosocial impairments, higher rates of traffic accidents, emergency department visits, poor quality of life, use of other drugs [16,17,18], cannabis withdrawal syndrome, and risk of addiction. Marijuana is additionally associated with increased risks of suicidal behavior and mania. With regard to depression and anxiety, the results are mixed, with some studies showing an association between marijuana and depression and anxiety [1,12,13], while other studies do not [14]. That the use of marijuana is a gateway to the use of other substances is of particular concern [6,16].

Next, due to structural and functional changes in brain morphology [19,20,21,22,23,24,25], the brains of adolescents and young adults are particularly vulnerable to the adverse effects of marijuana. Such vulnerability appears to be related to the greater sensitivity of the endocannabinoid system [26]. More specifically, Schonhofen et al. [27], in their overview, noted the importance of the endocannabinoid retrograde signaling pathway in regulating both excitatory and inhibitory synaptic plasticity via long-term potentiation and long-term depression. Cannabinoid receptors, endocannabinoids, and synthesis or degradation enzymes form the endocannabinoid system (ECS). This endocannabinoid system is functional from the early developmental stages and throughout adolescent cortical development. Schonhofen et al. [27] also observed that the endocannabinoid system, among others, regulates progenitor cell fate, neural differentiation, migration, and survival. Given this, it is not surprising that the endocannabinoid system may be particularly vulnerable to excessive cannabinoid exposure. Mandelbaum and de la Monte [28], in their critical review, commented that apart from anecdotal data and the high level of interest in the treatment of a broad range of illnesses, objective evidence concerning the short-term and long-term effects of continuous cannabis exposure on the developing brain remains limited. Mandelbaum and de la Monte [28] observed that the scarcity of long-term studies on the developing human brain is a matter for concern given that long-term exposure to cannabis negatively impacts on cognitive performance and particularly on white-matter brain tissue, where cannabinoid-1 receptors are abundant. 

To summarize, marijuana use has become a health issue both in Western countries and non-Western countries such as Iran, and this is particularly so for adolescents and young adults.

### 1.1. Theory of Planned Behavior (TPB)

Understanding the cognitive-emotional and behavioral factors underlying the intention to use marijuana is crucial to the effectiveness of countermeasures, such as preventive interventions to avoid or reduce its use. To this end, in the present study we applied the Theory of Planned Behavior (TPB) [29,30] for the following reasons. Firstly, this theory has already been successfully used to identify predictors of marijuana use among adolescents and young adults (see [31,32,33]). This theoretical model has thus proved to be applicable in explaining marijuana use on the behavioral level and in supporting specific psychotherapeutic interventions. However, secondly, cognitive-emotional and behavioral predictors of marijuana use have not, to our knowledge, been studied with respect to the Iranian population, and this holds particularly true for the TPB; yet this theory offers the possibility of specifying measures both to explain and to prevent marijuana use. Thus, we hoped in presenting this study to identify cognitive-emotional and behavioral factors with the capacity to improve both prevention and treatment.

Ajzen’s Theory of Planned Behavior (TPB) consists of six major components described in more detail below. The TPB claims that specific cognitive-emotional and attitudinal factors underlie the intention to perform a behavior. Intention is, in turn, a necessary condition for a particular behavior. Note that according to the theory, every behavior is preceded by an intention, but not every intention is necessarily translated into action.

Six cognitive-emotional factors impact on the nature and strength of an intention (here: “I want to smoke marijuana”) to perform a behavior (here: “I do smoke marijuana”). Personal skills in this context refer an individual’s problem-solving skills: “How do I solve problems? What can I do to solve a problem?” are typical questions related to personal skills. Environmental pressure or environmental circumstances refer to the possibility that the cognitive-emotional perception of an environmental context either hinders or facilitates access to an object and thus strengthens or reduces an intention: “How easy is it for me to obtain marijuana?” might be a typical question in the context of the present study. Self-efficacy refers to the belief that one can turn an idea or an intention into a successful achievement, and that such an achievement can be fully explained by one’s own engagement and performance. “I do not smoke marijuana because I avoid people smoking marijuana, and because I avoid places where marijuana is sold” might be the typical sentences of a person with high self-efficacy (to resist smoking marijuana). Attitude refers to an individual’s attitude towards an object or an action. “Marijuana is healthy!” might be a typical attitude towards the propensity to smoke marijuana. Subjective norms refer to the opinions existing in an individual’s social environment. “My friends find marijuana disgusting!” might be a typical statement describing a social environment hostile to its use.

### 1.2. Studies on Marijuana Use Based on the Theory of Planned Behavior

Malmberg et al. [31] applied the TPB in an investigation of the cognitive-behavioral predictors of intention and use of marijuana among young adolescents. They assessed 1023 Dutch adolescents aged 11 to 14 years. The sample completed a series of questionnaires assessing TPB dimensions. At this point, none of the participants were using marijuana. Twenty months later, they were assessed again. Results showed that a more positive attitude towards marijuana, greater perceived approval in the social environment, and lower self-efficacy to resist marijuana use predicted actual use via a stronger intention to use marijuana. Kam et al. [32] applied the TPB to predict alcohol, tobacco, and marijuana use among elementary school children in Mexico. At three different time points, participants completed questionnaires covering attitudes towards and actual use of alcohol, tobacco, and other drugs, along with questions on social rules. Results showed that supportive social norms (that is, parents and peers being perceived to have positive attitudes to use) impacted on the child’s own intentions, though this association was also mediated through a positive personal attitude towards alcohol, tobacco, and drug use and lower self-efficacy to resist their use. 

Lac et al. [34] investigated the predictors of marijuana use in a sample of 2141 adolescents aged 12 to 18 years using an extended TPB. They observed that greater parental knowledge of the health risks associated with marijuana use and parental warmth as a proxy for social norms predicted their adolescent children’s weaker pro-marijuana attitudes, and higher control over marijuana use as a proxy for self-efficacy. Additionally, these four factors reduced the intention to use marijuana. 

To summarize, the TPB and its extensions offer a valuable cognitive-emotional and environment-related framework for the explanation of behavioral intentions and behavior. Malmberg et al. [31], Kam et al. [32], and Lac et al. [34] have each successfully used different versions of TPB to predict marijuana use among adolescents and young adults. Given the lack of research with the TPB in Iran, the aim of the present study was to apply an extended TPB with respect to marijuana use among young adults in Iran, and to identify particular interventions that might aid both prevention and treatment.

Accordingly, and following Malmberg et al. [31], Kam et al. [32], and Lac et al. [34], we hypothesized that an extended TPB would have the capacity to predict both behavioral intentions and behavior with respect to marijuana use. Specifically, we expected that more supportive subjective social norms (towards marijuana use), poorer problem-solving skills, lower self-efficacy (to resist marijuana use), more positive attitudes towards marijuana use, and environmental circumstances perceived as favorable to marijuana use would predict a higher probability of actual marijuana use. 

To test this hypothesis, we assessed a sample of young Iranian adults. We believe that the study is of value for the following reasons. Firstly, marijuana and substance use is particularly high among young adults compared to adolescents and older adults in Iran [11]. Secondly, in so far as prevention programs focus on individuals’ cognitive-emotional processes, an extended version of TPB offers a variety of possible starting points for such programs. 

## 2. Methods

### 2.1. Procedure

Young adults in Esfahan and Kermanshah (Iran) who were then attending a walk-in center for drug use were approached via advertisements at universities and word-of-mouth recommendation with respect to potential participation in the present study. Eligible volunteers were fully informed about the aims of the study and the confidential handling of their data. Thereafter, they signed a written informed consent form. Next, they completed questionnaires covering sociodemographic information, marijuana use, and cognitive-emotional and behavioral dimensions related to marijuana use. Data were collected between January and March 2016. The Ethics Committee of the Kermanshah University of Medical Sciences (KUMS; Kermanshah, Iran) approved the study (IR.KUMS.REC.1398.1010), which was performed in accordance with the ethical principles laid down in the seventh and current edition (2013) of the Declaration of Helsinki.

### 2.2. Sample

A total of 180 participants were recruited. Inclusion criteria were: Being aged between 18 and 30 years; out-patients attending a walk-in center for drug use; reporting use of marijuana; being willing and able to comply with the study conditions, such as answering questions on marijuana use and completing questionnaires on cognitive-emotional and behavioral factors related to its use; and signed written informed consent. Exclusion criteria were: Having psychiatric diagnoses as their main diagnosis (e.g., major depressive disorders, bipolar disorders, posttraumatic stress disorder (PTSD), anxiety disorders, obsessive-compulsive disorders, personality disorders, eating disorders), as ascertained by an experienced clinical psychologist, and based on a brief neuropsychiatric interview (Mini Neuropsychiatric Interview [35]); primarily and regularly using other drugs, such as alcohol, opium- and opioid-containing medications, amphetamine, or methamphetamine, as ascertained by urine analysis; exclusion on the judgment of those running the study or experienced clinical psychologists that unreliable answers were given, either during the interview or in the completed questionnaires; and insufficient Farsi language skills. Tobacco use was not an exclusion criterion. 

Of the 180 individuals initially selected, 166 (92.2%) were included in the study. Six (3.3%) reported that drugs other than marijuana were their main focus; eight (4.4%) did not sign the written informed consent form. Table 1 provides participants’ sociodemographic information.

### 2.3. Tools

The questionnaire booklet had three sections: 1. Sociodemographic information; 2. Patterns of marijuana use; and 3. Questions covering TPB dimensions (see below for details). 

### 2.4. Socio-Demographic Characteristics

Participants provided information on the following: age (in years); gender (male or female); level of education (elementary school, secondary school, high school, university); marital status (single or married); parents’ current marital status (divorced: yes vs. no); parents’ education level (elementary school, secondary school, high school, university), and economic circumstances ((low = below USD 300); average (= between USD 301–700); good (= USD 701 and higher)) (see also Table 1).

### 2.5. Pattern of Marijuana Use

Participants were asked: “How many times have you used marijuana over the last week?”. 

### 2.6. Theory of Planned Behavior (TPB)

In the absence of a relevant Farsi questionnaire, items to assess TPB dimensions were taken from other scales [32,34]. To translate the items, we followed the algorithm proposed by Brislin [36]. Firstly, two English- and a Farsi-speaking translators independently translated the items. Secondly, the two versions of translated items were compared. In the case of complete linguistic and semantic overlap, the item remained unchanged. When linguistic and semantic overlap were low, a third translator endeavored find the best linguistic and semantic fit between divergent translations. Two independent translators then back-translated the Farsi version into English. In the case of high linguistic and semantic overlap between the original English items and the translated and back-translated version, the Farsi items were accepted as the final version. In the case of linguistic and semantic differences, both the Farsi and the translated English version were adapted until high linguistic and semantic overlap was achieved.

The final version of the questionnaire contained 33 items. Table 2 reports the items separately for the following dimensions: attitudes towards marijuana use; subjective norms; self-efficacy to resist marijuana use; environmental constraints; problem-solving skills; and behavioral intention with respect to use of marijuana.

The dimension attitude towards the marijuana use scale consisted of eight items. Typical items were: “Using marijuana….” “…is enjoyable”, “…improves my energy”, and “…improves my self-esteem”. Answers were given on seven-point Likert scales with the anchor point ranging from 1 (=strongly disagree) to 7 (=strongly agree), and with higher sum scores reflecting a more positive attitude towards marijuana use (Cronbach’s alpha = 0.94).

The subjective norm scale consisted of four items. Typical items were: “My friends encourage me to use marijuana”, and “When I use marijuana, I feel like I’m being accepted by my close friends”. Answers were given on five-point Likert scales with the anchor point ranging from 1 (=strongly disagree) to 5 (=strongly agree), and with higher sum scores reflecting subjective norms that are more supportive of marijuana use (Cronbach’s alpha = 0.86).

The scale assessing self-efficacy in resisting marijuana use consisted of seven items. Typical items were: “How sure are you to say no to marijuana in the following situations?”, “A close friend suggests you use marijuana”, “You are feeling sad”. Answers were given on five-point Likert scales with the anchor point ranging from 1 (=strongly disagree) to 5 (=strongly agree), and with higher sum scores reflecting greater self-efficacy in resisting marijuana use (Cronbach’s alpha = 0.84).

The scale assessing perceived environmental constraints on marijuana use consisted of eight items. Typical items were: “It is easy to get marijuana in this society”, and “I live in a neighborhood where using marijuana is normal”. Answers were given on five-point Likert scales with the anchor point ranging from 1 (=strongly disagree) to 5 (=strongly agree), and with higher sum scores reflecting perceived greater environmental constraints on marijuana use (Cronbach’s alpha = 0.72).

Self-perceived problem-solving was assessed with four items. Typical items were: “I think I have the ability to solve difficult problems”, and “I’m usually able to find creative solutions to solve a problem”. Answers were given on five-point Likert scales with the anchor point ranging from 1 (=strongly disagree) to 5 (=strongly agree), and with higher sum scores reflecting self-perceived greater problem-solving skills (Cronbach’s alpha = 0.72).

Behavioral intention with respect to marijuana use consisted of four items. Typical items were: “I intend to use marijuana in my life”, and “I intend to use marijuana in the next month.” Answers were given on five-point Likert scales with the anchor point ranging from 1 (=strongly disagree) to 5 (=strongly agree), with higher sum scores reflecting a stronger intention to use marijuana (Cronbach’s alpha = 0.77).

### 2.7. Statistical Analysis

Firstly, using a series of t-tests and ANOVAs, we tested whether marijuana use per week over the previous week differed systematically as a function of gender, educational level, parents’ marital status, parents’ educational level, or economic circumstances. Secondly, Pearson’s correlations were computed between the six dimensions of the extended TPB and reported frequency of marijuana use. Thirdly, a multiple linear regression analysis was executed to assess the predictors of marijuana usage. All statistical analyses were performed using SPSS 25.0 (IBM Corporation, Armonk, NY, USA). A p-value of less than 0.05 was considered as statistically significant.

## 3. Results

### 3.1. Sample Characteristics 

Participants’ mean age was 20.87 (SD = 1.80). All participants were single, and 26 (15.7%) were females. Table 1 summarizes the sociodemographic characteristics of the sample.

### 3.2. Mean Weekly Marijuana Use and Sociodemographic Dimensions

Mean frequency of use was 4.60 over the preceding week (SD = 3.17; range: 1–15).

Table 3 reports mean marijuana use as a function of gender, educational status, parents’ educational status, parents’ marital status, and economic circumstances. Males reported higher weekly marijuana use than females. Participants with divorced parents reported higher marijuana use per week than those with married parents. No differences were observed in weekly use as a function of participants’ educational level, parents’ educational level, or participants’ economic circumstances. 

### 3.3. Associations between Dimensions of the Extended Theory of Planned Behavior (TPB) and Weekly Marijuana Use

Table 4 gives the descriptive statistical indices (mean; standard deviation) and correlations between the six dimensions of the TPB (attitude towards marijuana use; subjective/social norms; environmental constraints; problem-solving skills, self-efficacy; intention to use marijuana) and weekly marijuana use.

A more positive attitude towards marijuana use (“marijuana is ‘good’”) was associated with subjective social norms more supportive of its use (“others also agree with marijuana use”), lower self-efficacy (“I’m less able to resist marijuana use”), stronger environmental constraints (“marijuana is readily available”), behavioral intention with respect to marijuana use (“I intend to use marijuana”), and weekly marijuana use. Attitude towards marijuana use was unrelated to problem-solving skills.

More supportive subjective social norms were associated with lower self-efficacy, higher environmental constraints, stronger behavioral intention with respect to marijuana use, and more frequent marijuana use. Subjective social norms were unrelated to problem-solving skills. 

Higher self-efficacy in resisting marijuana use was associated with lower environmental constraints, weaker behavioral intention with respect to marijuana use, and less frequent actual use. Self-efficacy in resisting marijuana use was unrelated to problem-solving skills. 

Higher environmental constraints were associated with poorer problem-solving skills, stronger behavioral intention with respect to marijuana use, and with more frequent actual use.

Better problem-solving skills were associated with a weaker behavioral intention with respect to marijuana use and with less frequent actual use.

A stronger behavioral intention to use marijuana use was associated with more frequent actual use.

### 3.4. Predicting Actual Use of Marijuana from Dimensions of the Extended Theory of Planned Behavior 

Table 5 reports the results of the multiple linear regression analysis, with the frequency of marijuana use as a dependent variable and the six dimensions of the extended TPB as predictors. These dimensions together predicted 33% of the variance in frequency of marijuana use. A positive attitude towards marijuana use, lower self-efficacy, and poorer problem-solving skills predicted more frequent marijuana use. Environmental constraints, subjective norms, and behavioral intention were excluded from the equation, as these dimensions did not achieve statistical significance. 

## 4. Discussion

The key findings of the present study were that, in a sample of young Iranian adults attending a walk-in center for drug use, dimensions of an extended Theory of Planned Behavior (TPB) and its facets were associated with the frequency of marijuana use. A set of cognitive-emotional and behavioral dimensions, including a positive attitude towards marijuana use, supportive subjective social norms with respect to marijuana use, limited self-efficacy to resist marijuana use, poor problem-solving skills, behavioral intention, and subjectively perceived low environmental constraints increased the intention to use marijuana, which in turn increased the odds of actually using marijuana. From a statistical point of view, a positive attitude towards marijuana use, low self-efficacy to resist its use, and poor problem-solving skills predicted more frequent actual use. The present results add to the current literature in an important way in that the Theory of Planned Behavior (TPB) identifies a range of cognitive-emotional factors that may explain use of marijuana, while the model also offers a theoretical foundation for practical and targeted interventions. 

Our hypothesis was that facets of the TPB could predict marijuana use. This hypothesis drew upon findings reported by Malmberg et al. [31], Kam et al. [32], and Lac et al. [34]. The hypothesis was supported. Accordingly, we conclude that the present pattern of results is consistent with previous efforts to apply the TPB in the explanation of marijuana use. The present findings expand upon previous research in that they were derived from a sample of young marijuana-using Iranian adults self-admitted to a daycare rehabilitation center for substance use disorders. 

In discussing the present findings, we focus on subjective social norms, self-efficacy, and problem-solving. In this respect, we understand the Discussion section as an attempt to embed the present results within a larger and more hypothetical framework. 

More supportive subjective social norms (i.e., “My peers or parents find marijuana use OK”) were associated with higher levels of marijuana use. This association reflects what Kam et al. [32] observed among a sample of Mexican children: the more parents (implicitly) consented to marijuana use, the greater were the odds that their children would use marijuana. In our view, this pattern of results reflects the close association between the set of (cognitive-emotional) behaviors present in the social environment and the individuals’ behavior. More specifically, the behavior of parents and peers are considered as legitimizing. While children have limited opportunities to choose their social environment—namely, their parents, siblings, social neighborhood, classmates, and sports mates, adults have more of a choice to actively accept or reject the members of their social environments. Following the seminal work of Festinger, Schachter, and Back [37], we suggest that participants in the present study may have actively chosen a peer group matching their own attitudes towards marijuana use so as to retain harmonious relations with their social environment. Festinger et al. [37] argued that individual beliefs, attitudes, or cognitions that differ from the beliefs, attitudes, and cognitions prevalent in their social environments by definition cause dissonance and discomfort. As Festinger and his colleagues anticipated, in order to reduce such discomfort and dissonance, individuals either change their attitudes and beliefs, or change their social environment. In the present study, participants had been self-admitted to a walk-in center for substance use to treat their marijuana use, and from this, the following two practical implications arose. Firstly, individuals’ motivation to change their behavior was high, and it was among the tasks of the walk-in center therapists to sustain these individuals’ motivation to change their behavior at the highest possible level. A second task for the therapists might have been to inform individuals about the basic processes outlined in Festinger et al.’s theory so as to encourage them to build alternative social environments, and more specifically, to modify their choice of peer group. 

With regard to self-efficacy, this concept derives from Bandura’s [38] seminal work on behavioral change. Briefly, self-efficacy refers to the individual’s capacity to plan their behavior, to turn plans into behavior, and to judge the success (or failure) of their efforts. In the present study, more limited self-efficacy in resisting marijuana use predicted a stronger intention to use marijuana and more frequent actual use. It follows that at a behavioral and interventional level, marijuana-using individuals should be encouraged to focus on occasions on which they successfully refrained from its use, and the cognitive-emotional and behavioral support they need to successfully resist marijuana use. We see this analysis as consistent with recent results. Pearson et al. [39] assessed a sample of college students (n = 1,123) and showed that sufficient levels of self-efficacy for marijuana refusal (along with marijuana-protective behavioral strategies) predicted lower levels of marijuana use. It follows that the individuals who had self-admitted themselves to a daycare center for substance use disorders to treat marijuana use should be instructed and reinforced in building-up strategies to resist marijuana use. Such strategies should include information about the negative long-term effects of marijuana [2,6,12,13,14,33,40,41,42], along with an explanation of the concept of self-control. Following cognitive-behavioral concepts [43], self-control is considered a specific behavior in a specific context which contains conflicting short-term and long-term outcomes. Self-control consists of two basic strategies: 1. Foregoing a positive reward in the *short-term* (here: the effect of marijuana) in order to achieve a positive outcome in the *long-term* (here: becoming psychologically and physiologically healthier; learning how to cope with (unpleasant) emotions; increasing self-efficacy); 2. Tolerating and dealing with unpleasant emotional states in the *short-term* (here: dealing with uncertainty, anxiety, or feelings of anger, sadness, frustration, or humiliation) in order to achieve a more stable state of self-regulation and cognitive-emotional independence from marijuana use in the *long-term.* In doing so, we also stress that substance use disorder is not considered a weakness of character, but rather a cognitive-emotional dilemma within a certain individual and environmental context and with respect to a specific issue. 

With regard to (the lack of) problem-solving skills, this is considered as mental flexibility in identifying adaptive solutions to problems encountered in everyday life [44]. Compared to individuals with no substance use disorder, individuals with substance use disorder reported a narrower range and poorer quality of problem-solving skills [44,45]. Specifically, Weiss et al. [45] showed that more effective problem-solving strategies, along with greater daytime use of distraction, and reappraisal predicted lower evening substance use. In contrast, higher evening substance use predicted higher next-day avoidance and reappraisal and poorer next-day problem-solving. In the context of the present study, psychological counseling might include teaching individuals, such as those in the present study, how to generate and explore behavioral alternatives to marijuana use. D’Zurilla and Golfried [46] proposed a structured seven-step model with regard to this: identify problems, define current status, describe aims, search for specific behaviors (and alternatives) to achieve aims, evaluate the efficacy of the procedures, and transfer the acquired problem-solving skills to other areas of everyday life, once the aim has been achieved (see D’Zurilla and Golfried [46] and [43] for a comprehensive description of this intervention). 

To summarize, results from the present study indicate that TPB offers a wide range of options for treating marijuana use at the practical level of interventions.

The results should be balanced against the limitations of the study. Firstly, we only assessed individuals attending a walk-in center for drug use, and for marijuana use more specifically. It follows that their responses cannot be compared with gender- and age-matched controls, or with gender- and age-matched individuals mainly using other drugs, such as amphetamine, methamphetamine, opium and opioid-containing medications, cocaine, or alcohol. Secondly, assessing a “pure” sample exclusively using marijuana does not reflect the reality of everyday clinical and psychiatric experience. Rather, individuals who are regular marijuana users often also report use of alcohol, opium, amphetamines, and other substances. Given that participants with psychiatric issues were excluded from the study, it is therefore possible that the self-efficacy responses regarding feeling sad and depressed might be biased. Thirdly, in a similar vein, behavioral disorders and attention bias were not assessed [47]. Fourthly, there was a lack of assessment on the impact of the media’s portrayal of medical marijuana and legalization of marijuana on positive attitudes of respondents in this study [48,49]. Fifthly, participants were assessed at the beginning of their treatment at the daycare rehabilitation centers, and it would have been interesting to investigate which of the six dimensions of the extended TPB changed after successful treatment, and which of the six dimensions predicted treatment success (or failure). Likewise, sixth, a follow-up assessment some months later would have allowed identification of predictors of treatment success or failure. Seventh, participants were all self-admitted to the walk-in center—by nature, self-referral is associated with insight into one’s problematic behavior. Again, it therefore follows that the present sample probably does not reflect the majority of young marijuana-using adults. Eighth, as shown in Table 1, the gender ratio was unbalanced—females were underrepresented, given that only about 25% of individuals attending the walk-in center were females. We have no explanation as to why less females volunteered to participate. Thus, the present findings might be biased, and future studies might seek to assess equal numbers of males and females. Lastly, assessing further samples, such as individuals with more severe addictions to marijuana, tobacco, alcohol, opioids, medication, or (meth-)amphetamines would have significantly enriched the present data and brought us to further conclusions.

## 5. Conclusions

Among a sample of young-adult marijuana users who were self-admitted to a walk-in center for drug use, facets of an extended Theory of Planned Behavior were found to be associated with more frequent marijuana use. From this study, we conclude that prevention programs to improve problem-solving skills and self-efficacy to resist marijuana use, along with educational work to highlight the negative effects of marijuana use might decrease such use among individuals in Iran.

## Figures and Tables

**Table 1 ijerph-17-01981-t001:** Sociodemographic information (N = 166).

Variables		M (SD)
Age (years)		20.87 (1.80)
	Number	Percent
Sex		
Female	26	15.7
Male	140	84.3
Education		
High school student (grade 7 to 11)	19	11.4
Diploma	33	19.9
College student	114	68.7
Father’s education		
Primary school	6	3.6
Diploma	65	39.2
Academic	95	57.2
Mother’s education		
Primary school	15	9
Diploma	74	44.6
Academic	77	46.4
Economic circumstances		
Low	11	6.6
Average	65	39.2
Good	90	54.2
Parents divorced		
Yes	17	10.2
No	149	89.8

**Table 2 ijerph-17-01981-t002:** Dimensions and descriptive statistical indices of the Theory of Planned Behavior.

Constructs	Mean	SD
Attitude towards Marijuana use Using marijuana……		
…. is enjoyable	5.13	2.12
…. is exciting	5.10	2.03
…. improves my energy	5.12	1.95
…. is attractive	5.07	1.96
…. is relaxing	5.09	2.10
…. improves my strength	4.69	1.81
…. improves my self-esteem	4.78	1.96
…. improves my mental abilities	4.97	1.90
Subjective norms		
My friends encourage me to use marijuana	3.03	0.94
When I use marijuana, I feel confirmed by my close friends	2.95	1.04
My place is encouraging to use marijuana	3.03	0.91
There is nothing wrong to me using marijuana	3.15	1.07
Self-efficacy to resist to marijuana use		
How likely are you to say “no” to Marijuana in the following situations?		
A close friend suggests you use marijuana	2.27	1.25
You are in a public place and someone offers marijuana to you	2.23	1.20
You are feeling happ.	2.37	1.19
You are feeling sad	2.34	1.15
You are feeling depressed	2.39	1.14
You are at a party where many people are using marijuana	2.37	1.35
You are feeling angry	2.66	1.13
Environmental constraints		
Our society has limited knowledge/education about the use of marijuana	2.74	1.20
There is a lack of available information about the side-effect of marijuana	2.63	1.11
It is easy to get marijuana in society	3.07	0.89
I have friends using marijuana	3.36	1.04
The police do not adequately supervise marijuana use and dealing	3.19	0.84
I’m living in a neighborhood where using marijuana is normal	3.16	0.95
Skills to solve problems		
I think I have the ability to solve difficult problems	3.09	1.08
I am usually able to find creative and effective alternatives to solve a problem	3.15	0.99
When I could not solve a problem, I analyze why it didn’t work	3.00	0.84
Before turning to action, I often consider a range of alternatives	2.95	0.93
Behavior Intention to using marijuana		
I intend to use marijuana in the next 6 months	3.27	0.97
I intend to use marijuana in the next 1 month	3.50	1.04
I intend to not use marijuana in my life	3.67	0.99
I intend to use marijuana when I am at a party	3.21	1.15

**Table 3 ijerph-17-01981-t003:** Association between sociodemographic variables and weekly marijuana use.

	Variable	n	Mean	SD	Statistics
Sex	Female	26	2.73	2.18	t(164) = 2.95, *p* = 0.004, d = 0.79
	Male	140	4.92	3.20	
Parents’ divorce	Yes	17	6.86	3.77	t(164) = 3.44, *p* = 0.001, d = 0.74
	No	149	4.32	2.98	
Education level	High school student	19	2.53	1.56	F(2, 163) = 0.93, *p* = 0.39
	Diploma	33	4.37	3.36	
	College student	114	4.97	3.18	
Fathers’ education level	Under diploma	6	6.50	2.88	F(2, 163) = 1.60, *p* = 0.19
	Diploma	65	4.51	3.67	
	Academic	95	4.52	2.83	
Mothers’ education level	Under diploma	15	6.16	4.44	F(2, 163) = 1.88, *p* = 0.14
	Diploma	74	4.05	2.71	
	Academic	77	4.77	3.19	
Economic status	Poor	11	4.00	1.69	F(2, 163) = 0.50, *p* = 0.61
	Average	65	4.55	3.20	
	Good	90	4.72	3.29	

**Table 4 ijerph-17-01981-t004:** Descriptive statistics and correlation coefficients of the six dimensions of the Theory of Planned Behavior (TPB) and the mean marijuana use per week.

Variables	Mean (SD)	Range	Attitude towards Marijuana Use	Subjective Norms	Self-Efficacy	Environmental Constraints	Skills to Solve Problems	Behavioral Intention for Marijuana Use	Marijuana Use/Week
Attitude towards marijuana use	36.98 (13.54)	8–56	-	0.29 **	−0.13	0.18 *	−0.09	0.31 **	0.50 **
Subjective norms	12.16 (3.41)	4–20		-	−0.18 *	0.21 **	0.00	0.42 **	0.40 **
Self-efficacy	16.67 (6.13)	7–35			-	−0.14	−0.09	−0.24 **	−0.24 **
Environmental constraints	18.16 (4.04)	6–30				-	−0.30 *	0.36 *	0.18 *
Skills to solve problems	12.24 (3.08)	4–20					-	−0.13	−0.18 *
Behavioral intention for marijuana use	13.62 (3.39)	4–20						-	0.39 **
Marijuana use per week	4.60 (3.17) ^1^	1–15							-

Notes: * *P* < 0.05, ** *P* < 0.01; ^1^ Kolmogorov–Smirnov test for normal distribution: *p* = 0.09.

**Table 5 ijerph-17-01981-t005:** Multiple linear regression with marijuana use per week, and the six dimensions of the extended Theory of Planned Behavior (TPB) (i.e., attitude towards marijuana use, subjective social norms, self-efficacy, environmental constraints, skill, intention to use marijuana) as predictors.

Dimension	Variables	Coefficient	Standard Error	Coefficient β	t	p	R	R^2^	Durbin-Watson Coefficient
Marijuana use per week	Intercept	0.22	1.76	-	12.94	0.00	0.60	0.34	1.63
	Attitude towards marijuana use	0.09	0.02	0.39	4.64	0.01			
	Self-efficacy	−0.09	0.04	−0.17	2.27	0.03			
	Skills to solve problems	−0.15	0.08	−0.15	1.99	0.04			
	Behavioral intention	0.19	0.09	0.19	2.11	0.04			
Excluded variables	Subjective norms	0.11	0.08	0.13	1.3	0.13			
	Environmental constraints	0.07	0.06	0.08	1.06	0.29			
	Intention	0.09	0.08	0.10	1.14	0.26			
	Gender^1^	0.28	0.19	0.18	1.47	0.09			
	Parents’ divorce	0.29	0.20	0.16	1.49	0.08			

Notes: Gender; 1 = males; 0 = females.

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
