# Peer review of "Extension of the Theory of Planned Behavior (TPB) to Predict Patterns of Marijuana Use among Young Iranian Adults"

_ijerph, 2020, doi:10.3390/ijerph17061981_

Round 1
Reviewer 1 Report
This study used Extension of the Theory of Planned Behavior (TPB) to predict patterns of marijuana use among Young Iranian Adults, which might be of significance among Iran young adults with a history of marijuana use. However, it has limited scientific contribution to understanding the behaviors of marijuana use since most of the findings are established.
The more serious issue is the methodological flaws: 1) I do not understand why a logistic regression model was used for a continuous outcome; 2) I saw little explanation of why the SEM was specified in the graph - seems it came from nowhere; 3) Textual description of results does not match the table results for the relationship between education and marijuana use; 4) discussion is well-written but is inadequately based on their findings - a lot of overstatement.
Author Response
Please see the detailed point-by-point-response.

Reviewer 2 Report
- In the Introduction, a brief description about the Iran’s government policy to marijuana use would be helpful for international reviewers to understand the background. Is marijuana use legal in Iran?
- A description about the major components or stages of the Theory of Planned Behavior is needed. What have been added in the extended version?
- The previous TPB studies can be summarized in one or two paragraphs instead of listing their details one by one.
- In the Methods, it is stated that subjects with mental health problems including depression and anxiety disorders were excluded. Given a high comorbidity of drug abuse with common mental disorders, what was the rationale for this exclusion criteria?
- The proportion of female participants was very low. How is it compared with the proportion of female users of marijuana in Iran?
- In the Results, Table 2 summarizes the participants’ attitudes towards marijuana use instead of sociodemographic (shown in Table 1).
- As the group of subjects with (diagnosed) mental health problems were excluded, the self-efficacy responses regarding feeling sad and depressed would be affected. This should be considered as a limitation of the study.
- Subjective norms, environment constraints and behavioral intention were excluded in multiple regression (Table 5), but they were included in the direct/indirect pathways (Table 6). It is quite confusing to the readers whether these factors were significant.
Author Response

(The authors gave the same response as above.)

Reviewer 3 Report
The theory of planned behavior model is a very powerful and predictive model for explaining human behavior. That is why the health field has been using this model often in their research studies. In current study, utilizing the theory of planned behavior, the researchers tried to predict patterns of marijuana use among young Iranian adults. The intention to predict risk factors in young people's behavior and prevent cannabis use was crucial in the research process. The presented topic indicates how important it is for educators to conduct appropriate public policy in order to prevent addictions.
The manuscript was written in a transparent manner, in accordance with the requirements of journal. However, the Authors did not avoid several shortcomings which, if corrected, will significantly increase the level of presented manuscript:
- Why were women only 15.7% of the surveyed? The number of people of both sexes should be similar in this type of research because it gives the opportunity to compare the results obtained depending on gender and draw conclusions e.g. which of group has a particular predisposition to specific behaviors and addictions.
- The Authors wrote that :”Due to structural and functional changes in brain morphology [18-24], the brains of adolescents and young adults are particularly vulnerable to the adverse effects of marijuana. Such vulnerability appears to be related to the greater sensitivity of the endocannabinoid system [25].” This is not entirely true. These changes are related among others to the period of use of stimulant and the number of cannabinoid receptors. This paragraph should be extended by the Authors
- The Authors claimed that participants of research currently attended a daycare rehabilitation center for drug use. does it mean that there were addicts? If so, then in this group the risk of behaviors leading to the abuse of various substances is greater than in the group of non-addicted people. Therefore, it is difficult to anticipate the impact of the environment and attitudes on the abuse of marijuana in these people.
- Line 225 it should be Table 2 instead of Table 1.
- Line 265 what kind of ANOVA was used? It should be included. Moreover, informations about the values of this analysis should be given. It is very important to estimate correctly the quality of the conducted considerations.
- Line 279 it should be Table 1 instead of Table 2.
- In my opinion, the averages in Table 2 do not present the correct picture about the behavior of young people, it should be presented in a different way. Perhaps rather as the number of people who chose the selected option.
- Line 291: The Authors claimed that” Participants with less education reported higher weekly marijuana use” however, this is not true when we look at the data in table 3. It should be corrected.
- In my opinion table 4 is not understandable. What does it represent? What correlations do the Authors point to? It should be explained.
- Line 357: „and its facets was able to predict frequency of marijuana” this is a far-reaching conclusion. We do not have any evidences that these behaviors were predicted in this way.
- Line 372-374 please correct the style of this sentence.
- Please explain the abbreviation SUD.
- Please rearrange the conclusion and improve the style of this paragraph.
- There are numerous punctuation errors and word aggregates in the manuscript.
- In my opinion the Authors should consider adding a group that is not placed in an addiction treatment center. Such comparison would significantly enrich the data presented by the Authors and perhaps bring them new conclusions.
Author Response

(The authors gave the same response as above.)

Round 2
Reviewer 1 Report
The authors have made significant improvements in their revised manuscript. However, I still have the following concerns about the statistical analyses, which I believe should be fully addressed before the manuscript is considered for publication.
1) Ideally, statistical analyses should facilitate hypothesis testing. In the introduction, the hypotheses were stated as follows: "Specifically, we expected that more supportive subjective social norms (towards marijuana use), poorer problem-solving skills, lower self-efficacy (to resist marijuana use), more positive attitudes towards marijuana use, and environmental circumstances perceived as favorable to marijuana use would predict both a stronger behavioral intention and a higher probability of actual marijuana use."
However, authors presented a Table 4 for demonstrating the correlations between 6 dimensions of TPB and marijuana use, and Table 5 for multiple linear regression for predicting marijuana use using the 6 dimensions. The first hypothesis regarding a stronger behavioral intention of marijuana use was not officially tested. The authors need to either rephrase their hypotheses, or run extra analyses with the behavioral intention as the outcome.
2) The authors should describe their rationale for specifying the multiple linear regression model for predicting marijuana use. Why those significant demographic variables such as sex and parents divorce were not included in the multivariable model building? Actually basic demographic variables such as age and education were usually included in such predictive model building since the p values can be misleading depending on the random sampling.
3) I was surprised that in the multiple linear regression results, behavioral intention was not a significant predictor for marijuana use, which is contradictory TPB. This makes me further question the model building process. Authors should either explain this results in their discussion, or take a closer loot at their model.
4) Based on the distribution of the marijuana use variable, mean = 4.6, SD = 3.17, range = 1 - 15, I suspect the distribution is not normal. Authors may consider presenting skewness indicator. If the normality error assumption is seriously violated, t-test and ANOVA are not appropriate in testing differences for variables in Table 1.
Author Response
Thank you; see the detailed point-by-point-response.

Reviewer 2 Report
The authors have revised the paper according to most of the comments from the reviewers. I am satisfied with the revision and responses.
Author Response

(The authors gave the same response as above.)
